# Effect of Significant Coronary Artery Stenosis on Prognosis in Patients with Vasospastic Angina: A Propensity Score-Matched Analysis

**DOI:** 10.3390/jcm10153341

**Published:** 2021-07-28

**Authors:** Hyun-Jin Kim, Min-Ho Lee, Sang-Ho Jo, Won-Woo Seo, Hack-Lyoung Kim, Kwan-Yong Lee, Tae-Hyun Yang, Sung-Ho Her, Seung-Hwan Han, Byoung-Kwon Lee, Keun-Ho Park, Seung-Woon Rha, Hyeon-Cheol Gwon, Dong-Ju Choi, Sang-Hong Baek

**Affiliations:** 1Division of Cardiology, Department of Internal Medicine, Hanyang University College of Medicine, Seoul 04763, Korea; titi8th@gmail.com; 2Division of Cardiology, Department of Internal Medicine, Soonchunhyang University Seoul Hospital, Seoul 04401, Korea; neoich@gmail.com; 3Cardiovascular Center, Hallym University Sacred Heart Hospital, Anyang-si 14068, Korea; 4Division of Cardiology, Department of Internal Medicine, Kangdong Sacred Heart Hospital, Hallym University College of Medicine, Seoul 05355, Korea; wonwooda@gmail.com; 5Cardiovascular Center, Seoul National University Boramae Medical Center, Seoul 07061, Korea; khl2876@gmail.com; 6Department of Cardiovascular Medicine, Incheon St. Mary’s Hospital, The Catholic University of Korea, Incheon 21431, Korea; kyle210@naver.com; 7Department of Cardiovascular Medicine, Busan Paik Hospital, Inje University, Busan 47392, Korea; yangthmd@naver.com; 8Department of Cardiovascular Medicine, St. Vincent’s Hospital, The Catholic University of Korea, Suwon 16247, Korea; hhhsungho@hanmail.net; 9Department of Cardiovascular Medicine, Gil Medical Center, Gachon University, Incheon 21565, Korea; shhan@gilhospital.com; 10Department of Cardiovascular Medicine, Gangnam Severance Hospital, Yonsei University, Seoul 06273, Korea; cardiobk@gmail.com; 11The Heart Center, Chosun Medical Center, Gwangju 61453, Korea; keuno21@naver.com; 12Department of Cardiovascular Medicine, Guro Hospital, Korea University, Seoul 08308, Korea; swrha617@yahoo.co.kr; 13Department of Cardiovascular Medicine, Samsung Medical Center, Sungkyunkwan University, Seoul 06351, Korea; hcgwon@naver.com; 14Division of Cardiology, Department of Internal Medicine, Seoul National University Bundang Hospital, Seongnam 13620, Korea; djchoi@snubh.org; 15Division of Cardiology, Seoul St. Mary’s Hospital, The Catholic University of Korea, Seoul 06649, Korea; whitesh@catholic.ac.kr

**Keywords:** vasospastic angina, coronary artery stenosis, acute coronary syndrome

## Abstract

Vasospastic angina (VA) is a functional disease of the coronary artery and occurs in an angiographically normal coronary artery. However, it may also occur with coronary artery stenosis. We investigated the effect of coronary artery stenosis on clinical outcomes in VA patients. Study data were obtained from a prospective multicenter registry that included patients who had symptoms of VA. Patients were classified into two groups according to presence of significant coronary artery stenosis. Among 1920 patients with VA, 189 patients were classified in the “significant stenosis” group. The one-year composite clinical events rate was significantly higher in the significant stenosis group than in the “no significant stenosis” group (5.8% vs. 1.4%, respectively; *p* < 0.001). Additionally, the prevalence of ACS was significantly greater in the “significant stenosis” group (4.8% vs. 0.9%, respectively; *p* < 0.001). After propensity score matching, the adverse effects of significant stenosis remained. In addition, significant stenosis was independently associated with a 6.67-fold increased risk of ACS in VA patients. In conclusion, significant coronary artery stenosis can increase the adverse clinical outcomes in VA patients at long-term follow-up. Clinicians should manage traditional risk factors associated with atherosclerosis and control vasospasm as well as reduce the burden of atherosclerosis.

## 1. Introduction

Vasospastic angina (VA) is caused by focal or diffuse spasm of an epicardial coronary artery, resulting in severe obstruction of the coronary artery lumen and myocardial ischemia [1]. Vasospasm can occur in an angiographically normal coronary artery, but may also occur at the site of an existing organic atherosclerotic stenosis [2]. Stable atherosclerotic plaques are rarely fatal, but can interfere with coronary blood flow and lead to stable angina [3]. However, it has been suggested that vasospasm is associated with endothelial damage and subsequent atheroma rupture [4]. Considering that acute coronary syndrome (ACS) is almost always caused by luminal thrombus or sudden plaque rupture applied to organic atherosclerotic plaques [5], coronary vasospasm can induce the rupture of a stable atheroma, which could lead to myocardial infarction and sudden cardiac death.

Overall, VA has a good long-term prognosis [6]. On the contrary, previous small studies have shown that significant coronary artery atherosclerotic stenosis is associated with a worse clinical outcome in patients with VA [7,8,9]. In other studies, there was no significant difference in prognosis of VA patients with or without significant stenosis [10]. Notably, most of the above studies showed the clinical outcome only in patients with VA, excluding patients with significant coronary artery stenosis [11,12,13]. Few studies have directly compared and evaluated in detail the clinical prognosis of vasospasm in patients with or without significant coronary stenosis with a long-term follow-up. Therefore, we investigated the effect of significant coronary artery stenosis on clinical outcomes in VA patients using a large-scale nationwide prospective registry.

## 2. Materials and Methods

### 2.1. Study Population

Study data were obtained from a nationwide prospective Vasospastic Angina in Korea (VA-Korea) registry. The study design of VA-Korea has been published previously [6,14,15]. Eleven tertiary hospitals in Korea participated in this registry between May 2010 and June 2015. Patients over 18 years old with suspected symptoms of VA who underwent invasive coronary angiography (CAG) with ergonovine (EG) provocation test were consecutively entered into the registry. Of the 2960 initially enrolled patients with suspected VA, 1987 patients had intermediate spasm or significant spasm after intracoronary EG injection during CAG. Of these, 67 patients were excluded because they were lost to follow-up. Thus, the data from 1920 patients with VA were used for the final analysis. Among them, 1731 patients were classified into the “no significant stenosis” group (<50% luminal diameter narrowing of coronary arteries) and 189 patients into the “significant stenosis” group (≥50% luminal diameter narrowing of one or more coronary arteries). This study protocol complied with the Declaration of Helsinki and was reviewed and approved by the Institutional Review Board of Hallym University Sacred Heart Hospital (Approval No. 2010-I007). All patients provided written informed consent prior to study entry.

### 2.2. Data Collection

The patient data were obtained from the VA-Korea database via a web-based electronic data capture system including an electronic case report form. The following patient demographic and clinical characteristics were extracted from this database: age, sex, body mass index (BMI; kg/m^2^), blood pressure, previous history of coronary artery disease, diabetes mellitus, hypertension, dyslipidemia, alcohol drinking status, current smoking status, and previous cardiovascular medications. These previous histories were obtained by reviewing each patient’s medical history. Laboratory data were also collected: hemoglobin, creatinine, glucose, high-sensitivity C-reactive protein, total cholesterol, low-density lipoprotein cholesterol, triglycerides, and high-density lipoprotein cholesterol. Left ventricular ejection fraction from echocardiography data upon admission was also collected.

### 2.3. Invasive CAG and EG Provocation Test

Baseline CAG and EG provocation tests were performed on patients who had suspicious symptoms of VA at the discretion of the clinician in accordance with the Guidelines for Diagnosis and Treatment of Patients with VA of the Japanese Circulation Society [1]. The baseline CAG was performed by a well-trained cardiologist; vasoactive medications were stopped at least 48 h before the procedure. For provocation testing, intracoronary EG was injected into the left coronary artery (LCA) in incremental doses of 20 (E1), 40 (E2), and 60 (E3) μg. If coronary spasm was not induced in the LCA, incremental doses of 10 (E1), 20 (E2), and 40 (E3) μg were injected into the right coronary artery (RCA) [1,16]. When coronary spasm was induced, 200 μg of nitroglycerine was injected. During the provocation test, the location of spasm, presence of chest pain, and electrocardiography (ECG) changes were recorded. Definition of ECG change was ST-segment elevation or depression (≥1 mm) or T-wave inversion in at least 2 consecutive leads. Significant vasospasm was defined as a total or luminal diameter narrowing, by more than 90%, of the coronary arteries accompanying chest pain and/or ECG changes after EG injection [1]. The definition of intermediate vasospasm was 50% to 90% luminal diameter narrowing of the coronary arteries. All patients who had spontaneous vasospasm or vasospasms on the EG provocation test were treated with angina medication according to the clinician’s discretion during follow-up.

### 2.4. Study Outcomes

The primary outcome was the composite clinical events rate during one year of follow-up (median duration, 359 days; mean 285.8 ± 129.3 days). Composite clinical events included cardiac death, new-onset arrhythmia including ventricular tachycardia (VT) and ventricular fibrillation (VF), and atrioventricular (AV) block. VT was defined as sustained VT leading to hemodynamic instability, whereas AV block was defined as high-degree AV block leading to hemodynamic instability. Deaths from all causes were also observed during the one-year follow-up. In addition, data on composite clinical events and all-cause death during long-term follow-up (median duration, 757.5 days; mean 723.5 ± 482.0 days) were collected. The occurrence and timing of death were investigated with the review of medical records or from a telephone interview.

### 2.5. Statistical Analyses

All categorical data are expressed as frequencies and percentages, and continuous variables are shown as means and standard deviations. Pearson’s chi-squared test was used to compare categorical variables. For continuous variables, the Shapiro–Wilk test was used for confirming the normal distribution of each dataset. Then, the student’s *t*-test was used to compare normally distributed continuous variables, and the Mann–Whitney U test was used to compare non-normally distributed continuous variables. In addition, we adjusted the uneven distribution of baseline characteristics between the “significant stenosis” group and the “no significant stenosis” group by using a propensity score and 1:2 matched analysis. A multiple logistic regression analysis model was built to represent the propensity score, which was the probability of the nitrates group. The adjusted variables were age, sex, systolic blood pressure, diastolic blood pressure, BMI, history of coronary artery disease, hypertension, diabetes, dyslipidemia, alcohol drinking, and current smoking status. The 182 patients in the “significant stenosis” group were matched to 364 patients in the “no significant stenosis” group. McNemar’s test was used to compare categorical variables between the matched patient groups, and a paired *t*-test was used for continuous variables. Kaplan–Meier survival analysis and log-rank test were used to compare ACS-free survival rates and cumulative composite clinical events-free survival rates between the two groups. Univariate analysis and subsequent multivariate logistic regression analysis were also performed to evaluate the risk of ACS after adjustment for individual risk factors. Age, sex, and variables with predictive significance (*p* < 0.05) of ACS in univariate analysis were included in the regression analysis. A *p*-value less than 0.05 was considered statistically significant. All analyses were performed using SPSS v.21.0 (IBM Corp., Armonk, NY, USA).

## 3. Results

### 3.1. Baseline Characteristics

Among 1920 patients with VA including intermediate and significant spasm, there were 1731 patients who had no significant coronary artery stenosis and 189 patients who had significant coronary artery stenosis at baseline CAG. Out of 189 patients, there were 115 patients with significant stenosis in the left anterior descending coronary artery, 37 patients with significant stenosis in the left circumflex coronary artery, and 70 patients with significant stenosis in the right coronary artery. There was no patient who had significant stenosis of the left main coronary artery. Patients’ baseline characteristics according to presence of coronary artery stenosis are shown in Table 1. Patients in the “significant stenosis” group were significantly older, were more likely to be male, had more previous diabetes mellitus and hypertension, and reported more alcohol drinking and current smoking than those in the “no significant stenosis” group. There were no significant differences in other histories or medications related to traditional cardiovascular risk factors or diseases between the two groups.

### 3.2. Clinical Outcomes According to Significant Coronary Artery Stenosis

Among 1920 patients, the composite clinical events of ACS, cardiac death, VT or VF, or AV block occurred in 36 patients (1.9%) during the 1-year follow-up. As shown in Table 2, the 1-year composite clinical events rate was significantly higher in the “significant stenosis” group than in the “no significant stenosis” group (5.8% vs. 1.4%, respectively; *p* < 0.001). Specifically, the prevalence of 1-year ACS was significantly more frequent in the “significant stenosis” group than in the “no significant stenosis” group (4.8% vs. 0.9%, respectively; *p* < 0.001). However, 1-year all-cause death rates did not differ. Based on whether the VA patients had significant coronary artery stenosis, the cumulative composite clinical events rate and the cumulative ACS-free survival rate were analyzed, and results are shown in Figure 1A,B.

Patients in the “significant stenosis” group had a significantly lower cumulative death-free survival rate than patients in the “no significant stenosis” group at 1-year follow-up (92.8% vs. 98.1%, respectively; log-rank *p* < 0.001) (Figure 1A). Patients in the “significant stenosis” group also had a significantly lower cumulative ACS-free survival rate (95.2% vs. 99.1%, respectively; log-rank *p* < 0.001) (Figure 1B).

In addition, the composite clinical events occurred in 75 patients (3.9%) during follow-up and 12 patients (0.6%) died from all-cause during long-term follow-up (median duration, 757.5 days; mean 723.5 ± 482.0 days) (Table 3). Additionally, the prevalence of composite clinical events was more frequent in the “significant stenosis” group than in the “no significant stenosis” group during long-term follow-up. The prevalence of ACS was more frequent in the “significant stenosis” group, but there was no statistically significant difference in all-cause death rates between the two groups.

Appendix A shows the subgroup analysis conducted to determine the difference in clinical events depending on one-vessel disease or multi-vessel disease (stenosis of two or more coronary arteries). Among 189 VA patients with significant stenosis, 161 patients had one-vessel disease and 28 patients had multi-vessel disease. There was no significant difference in composite clinical events, ACS, or all-cause death rates between the one-vessel disease group and multi-vessel disease group during 1-year follow-up as well as long-term follow-up.

### 3.3. Clinical Outcomes in Propensity Score-Matched Population

After propensity score matching, 182 patients in the “significant stenosis” group were successfully matched to 364 patients in the “no significant stenosis” group. Baseline characteristics were not significantly different between groups after propensity score matching (Appendix A). The 1-year composite clinical events rate of the matched population was significantly higher in the “significant stenosis” group (11 patients among 182 patients (7.4%) vs. 8 patients among 364 patients (3.0%), respectively; *p* = 0.035) (Table 4). In addition, the 1-year ACS events rate of the matched population was significantly higher in the “significant stenosis” group than the “no significant stenosis” group (9 patients among 182 patients (4.9%) vs. 5 patients among 364 patients (1.4%), respectively; *p* = 0.019). Furthermore, the prevalence of composite clinical events and ACS was also more frequent in the “significant stenosis” group of the matched population during long-term follow-up. Figure 2 shows the cumulative composite clinical events rate and cumulative ACS-free survival rate between the matched groups. Patients in the “significant stenosis” group had a lower cumulative event-free survival rate than patients in the “no significant stenosis” group (94.0% vs. 97.8%, respectively; log-rank *p* = 0.051) (Figure 2A), and had a statistically significant lower cumulative ACS-free survival rate (95.1% vs. 98.6%, respectively; log-rank *p* = 0.029) (Figure 2B).

### 3.4. Effect of Significant Coronary Artery Stenosis on 1-Year ACS Rate in VA Patients

According to univariate analysis (Table 5), the following factors were associated with ACS events at 1-year follow-up in VA patients: significant stenosis (odds ratio [OR], 5.36; 95% confidence interval [CI], 2.335–12.303; *p <* 0.001) and dyslipidemia. After adjusting for age, sex, and dyslipidemia, the Cox regression analysis showed that significant stenosis was independently associated with a 6.67-fold increased hazard for ACS in VA patients (OR, 6.67; 95% CI, 2.798–15.908; *p* < 0.001). Dyslipidemia was also independently associated with ACS events at 1-year follow-up.

## 4. Discussion

According to results from this nationwide prospective large-scale registry, the incidence of 1-year composite clinical events including ACS was significantly higher in VA patients who had significant coronary artery stenosis at baseline CAG than those who had no significant stenosis; the adverse effects of significant stenosis were consistent after propensity score matching. However, there was no difference in clinical events between one-vessel coronary disease and multi-vessel disease. Especially, significant coronary artery stenosis was independently associated with a 6.67-fold increased risk of ACS in patients with VA at 1-year follow-up. Dyslipidemia, a traditional cardiovascular risk factor, was also independently associated with an increased risk of ACS.

Vasospasm is common in patients with no or mild coronary artery stenosis [17]. Although the tendency of coronary artery spasm may be the only cause of functional coronary artery abnormalities, it also overlaps with significant coronary artery stenosis. However, the prevalence of co-existing significant coronary artery stenosis in VA patients was reported to be relatively low, within 10% in a previous Japanese study [8,18]. In our study, 9.8% of patients with VA also had a significant stenosis rate; this rate is comparable to the previous reports. Despite a low incidence of significant atherosclerotic coronary stenosis in VA patients, once an atherosclerotic plaque is present, it can contribute to the development of ACS by coronary vasospasm [19,20]. Ishii M. et al. [21] showed the clinical outcome of patients with coronary spasm combined with significant atherosclerotic stenosis. Among 1760 patients with typical or atypical angina-like chest pain who underwent provocation test, 358 (20.3%) patients had significant stenosis. Of the 358 patients with significant stenosis, 233 (65.1%) patients showed vasospasm after provocation test. Contrary to the design of our study, they demonstrated that provoked spasm at the site of significant stenosis was an independent risk factor for major adverse cardiac events. Those mechanisms have not been clearly demonstrated in human studies; an animal study showed that vasospasm might be one of the mechanisms triggering atherosclerotic plaque injury and subsequent acute ischemic myocardial injury [22]. Coronary artery stenosis and subsequent plaque rupture with thrombus formation after vasospasm in animal models have many obvious differences with the human and clinical situation. However, this mechanism could provide theoretical support to the significantly greater incidence of clinical events, including ACS, in VA patients with significant stenosis compared to VA patients without significant stenosis in our study.

In the present study, the multivariate analysis also demonstrated that significant coronary artery stenosis was a strong and significantly correlated factor of 1-year ACS in patients with VA by 6.67-fold. Takagi et al. [8] also showed that significant coronary artery stenosis (≥50% luminal diameter narrowing) independently increased the risk of major adverse cardiac events by 2.04-fold in VA patients who survived out-of-hospital cardiac arrest during a 32-month follow-up period. Their study targeted higher acuity patients (who had cardiac arrest) compared with our study, and they included additional major adverse cardiac events such as cardiac death, nonfatal myocardial infarction, hospitalization for unstable angina and heart failure, and appropriate ICD shocks. Their findings are comparable with our study in that significant stenosis can increase cardiac events, but the more severe characteristics of the enrolled patients and diverse clinical outcomes differed from our study. In another study, Takatsu et al. [23] showed that mild coronary artery stenosis above 0% increased the risk of major adverse cardiac events including acute myocardial infarction, unstable angina, and development of severe coronary disease by 1.66-fold over an 11-year follow-up. While they enrolled patients excluding organic significant stenosis, which was different from our study, their findings highlight the importance of coronary artery stenosis at baseline CAG in patients with VA by demonstrating that even a small atherosclerotic burden can contribute to the increase in adverse cardiac events.

Dyslipidemia, as a traditional cardiovascular risk factor of atherosclerotic cardiovascular disease [24], was also an independent risk factor for 1-year ACS in patients with VA regardless of significant coronary artery stenosis at baseline CAG. Although VA is a functional disease [1], the traditional cardiovascular risk factors should be remedied through appropriate medical therapy.

There are few studies with direct comparisons between VA patients with or without significant coronary stenosis and detailed evaluations of the clinical prognosis over a long-term follow-up. We presented refined data results by adjusting the baseline characteristics of VA patients with or without significant coronary stenosis by performing propensity-score matching. Therefore, our study is novel and more rigorous compared with previous studies in that no previous studies have performed analysis on a matched population.

Several limitations of this study must be considered. First, this was a prospective multicenter cohort study and may have unavoidable methodological biases that could impact the results. However, to reduce bias as much as possible, we performed propensity score matching and multivariate logistic regression. Second, although stenosis and vasospasm of coronary artery were evaluated in this VA-Korea registry, it was not possible to determine whether vasospasm occurred in the presence of atherosclerotic stenosis. However, it yielded a meaningful finding that suggested VA patients with significant stenosis had worse clinical outcomes, whether the vasospasm occurred in the fixed atherosclerotic lesion or caused an atherosclerotic burden in another, non-spastic coronary artery. Third, there was no information on whether coronary intervention had been performed in the “significant stenosis” group after baseline CAG, which could affect the clinical outcome in VA patients. In addition, there was no information on drug therapy after vasospasm was demonstrated by EG provocative tests, which could also have affected the clinical prognosis in VA patients. However, although drug information was limited and not presented here, there was no significant difference between the “significant stenosis” group and “no significant stenosis” group regarding whether drug therapy was maintained during the follow-up period.

## 5. Conclusions

In conclusion, VA patients with significant coronary artery stenosis had significantly worse clinical outcomes, including ACS, during the follow-up period; this was irrespective of whether their stenosis was a one-vessel or multi-vessel disease. Thus, in the management of VA patients, clinicians should pay attention to and manage traditional risk factors associated with atherosclerosis, control vasospasm, and reduce the burden of atherosclerosis in order to achieve better clinical outcomes.

## Figures and Tables

**Figure 1 jcm-10-03341-f001:**
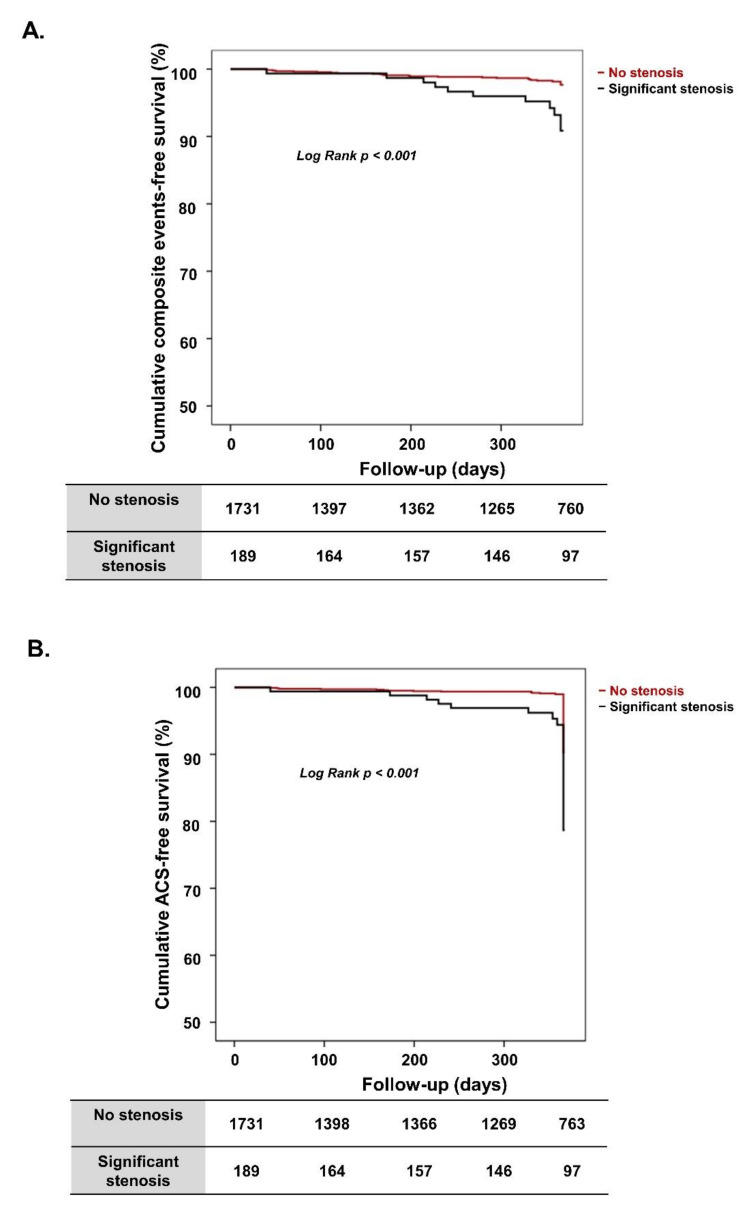
Kaplan–Meier survival curves for the entire population. (**A**) Cumulative composite event-free survival according to presence of significant stenosis. (**B**) Cumulative ACS-free survival according to presence of significant stenosis. ACS, acute coronary syndrome.

**Figure 2 jcm-10-03341-f002:**
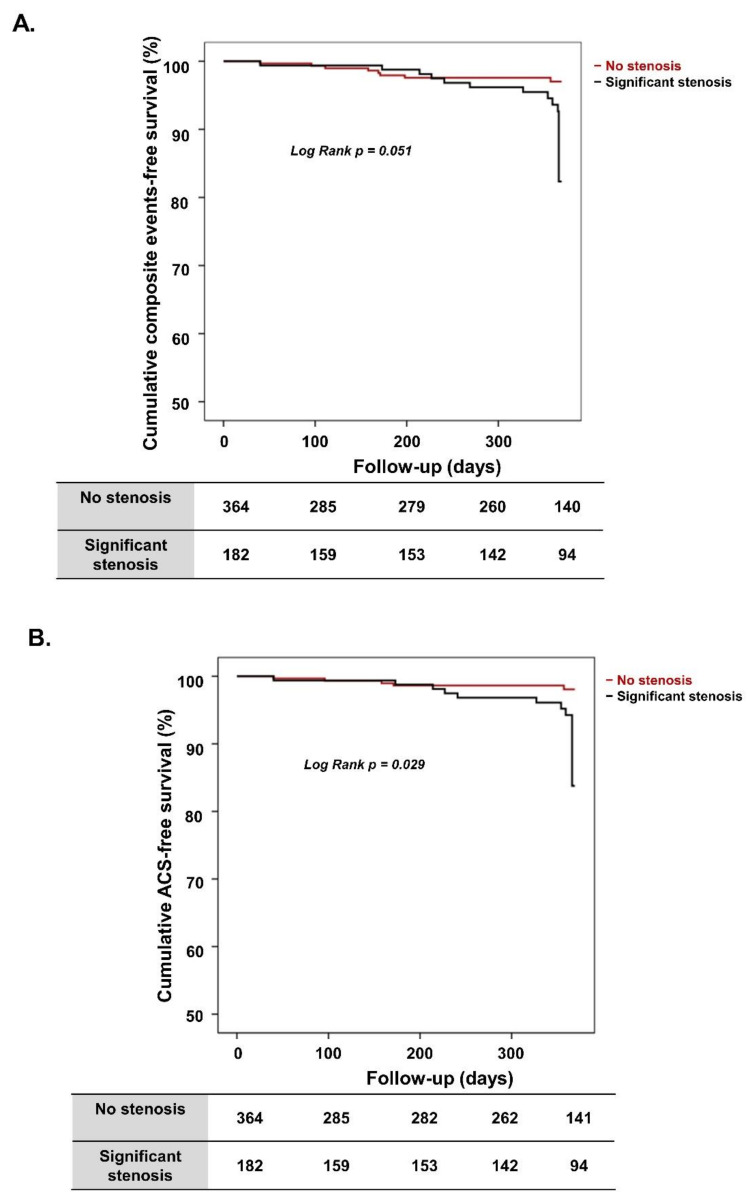
Kaplan–Meier survival curves in propensity score-matched population. (**A**) Cumulative composite event-free survival according to presence of significant stenosis. (**B**) Cumulative ACS-free survival according to presence of significant stenosis. ACS, acute coronary syndrome.

**Table 1 jcm-10-03341-t001:** Baseline characteristics.

	All (*n* = 1920)	No Significant Stenosis (*n* = 1731)	Significant Stenosis (*n* = 189)	*p*-Value
Age, years	55.1 ± 11.3	54.7 ± 11.3	58.7 ± 10.6	<0.001
Male, *n* (%)	1185 (61.7)	1046 (60.4)	139 (73.5)	<0.001
BMI, kg/m^2^	24.7 ± 3.4	24.8 ± 3.3	24.7 ± 3.8	0.676
SBP, mmHg	126.2 ± 18.4	126.1 ± 18.2	127.0 ± 19.7	0.506
DBP, mmHg	77.0 ± 12.3	77.0 ± 12.1	76.3 ± 13.8	0.427
Previous CAD *, *n* (%)	247 (12.9)	215 (12.4)	32 (16.9)	0.080
Diabetes mellitus, *n* (%)	180 (9.4)	146 (8.4)	34 (18.0)	<0.001
Hypertension, *n* (%)	731 (73.1)	9\636 (36.8)	95 (50.3)	<0.001
Dyslipidemia, *n* (%)	313 (16.4)	283 (16.4)	30 (15.9)	0.848
Alcohol drinking, *n* (%)	794 (41.4)	702 (40.6)	92 (48.7)	0.031
Current smoking, *n* (%)	524 (27.7)	458 (26.8)	66 (35.3)	0.014
Laboratory findings				
Hemoglobin, g/dL	13.9 ± 1.9	13.9 ± 1.9	13.9 ± 1.5	0.949
Creatinine, mg/dL	0.8 ± 0.4	0.8 ± 0.4	0.8 ± 0.2	0.593
Glucose, mg/dL	111.4 ± 37.7	111.3 ± 38.3	113.0 ± 31.9	0.573
hs-CRP, mg/dL	0.7 ± 4.4	0.7 ± 4.5	0.6 ± 3.1	0.810
Total cholesterol, mg/dL	174.2 ± 36.3	174.8 ± 36.0	168.5 ± 38.6	0.029
LDL cholesterol, mg/dL	103.7 ± 31.5	104.3 ± 31.3	98.6 ± 33.5	0.030
Triglyceride, mg/dL	141.9 ± 104.3	140.2 ± 103.2	157.0 ± 113.0	0.049
HDL cholesterol, mg/dL	46.8 ± 12.8	47.1 ± 12.9	44.1 ± 11.2	0.001
LV EF, %	64.5 ± 6.6	64.6 ± 6.4	63.5 ± 7.8	0.108
Previous cardiovascular medications				
Antiplatelet, *n* (%)	424 (22.1)	371 (21.4)	53 (28.0)	0.060
Statin, *n* (%)	296 (15.4)	258 (14.9)	38 (20.1)	0.132
CCB, *n* (%)	375 (19.5)	327 (18.9)	48 (25.4)	0.075
Clinical diagnosis before ergonovine				
Angina, *n* (%)	1740 (90.6)	1572 (90.8)	168 (88.9)	0.388
Myocardial infarction, *n* (%)	35 (1.8)	29 (1.7)	6 (3.2)	0.146
Cardiac arrest, *n* (%)	27 (1.4)	25 (1.4)	2 (1.1)	1.000
Syncope, *n* (%)	24 (1.3)	22 (1.3)	2 (1.1)	1.000
VT or VF, *n* (%)	12 (0.6)	11 (0.6)	1 (0.5)	1.000
AV block, *n* (%)	1 (0.1)	1 (0.1)	0 (0.0)	1.000

AV, atrioventricular; BMI, body mass index; CAD, coronary artery disease; CCB, calcium-channel blocker; DBP, diastolic blood pressure; HDL, high-density lipoprotein; hs-CRP, high sensitive-C reactive protein; LDL, low-density lipoprotein; LV EF, left ventricular ejection fraction; SBP, systolic blood pressure; VF, ventricular fibrillation; VT, ventricular tachycardia. * Previous CAD included angina with evidence of ischemic heart disease, CAD with medical treatment after CAG, CAD with percutaneous coronary intervention, and CAD with coronary artery bypass graft.

**Table 2 jcm-10-03341-t002:** One-year clinical event rate for patients with VA according to significant stenosis of coronary artery.

	All (*n* = 1920)	No Significant Stenosis (*n* =1731)	Significant Stenosis (*n* = 189)	*p*-Value
Composite events	36 (1.9)	25 (1.4)	11 (5.8)	<0.001
ACS	25 (1.3)	16 (0.9)	9 (4.8)	<0.001
Cardiac death	2 (0.1)	2 (0.1)	0 (0.0)	1.000
VT or VF	3 (0.2)	2 (0.1)	1 (0.5)	0.267
AV block	6 (0.3)	5 (0.3)	1 (0.5)	0.463
All-cause death	8 (0.4)	8 (0.5)	0 (0.0)	1.000

ACS, acute coronary syndrome; AV, atrioventricular; VA, vasospastic angina; VF, ventricular fibrillation; VT, ventricular tachycardia.

**Table 3 jcm-10-03341-t003:** Clinical event rate of patients with VA according to significant stenosis of coronary artery.

	All (*n* = 1920)	No Significant Stenosis (*n* =1731)	Significant Stenosis (*n* = 189)	*p*-Value
Composite events	75 (3.9)	58 (3.4)	17 (9.0)	0.001
ACS	59 (3.1)	45 (2.6)	14 (7.4)	0.001
Cardiac death	3 (0.2)	3 (0.2)	0 (0.0)	1.000
VT or VF	8 (0.4)	6 (0.3)	2 (1.1)	0.182
AV block	7 (0.4)	6 (0.3)	1 (0.5)	0.516
All-cause death	12 (0.6)	12 (0.7)	0 (0.0)	0.621

ACS, acute coronary syndrome; AV, atrioventricular; VA, vasospastic angina; VF, ventricular fibrillation; VT, ventricular tachycardia.

**Table 4 jcm-10-03341-t004:** Clinical event rate of patients with VA according to significant stenosis of coronary artery in matched population.

One-Year	All (*n* = 546)	No Significant Stenosis (*n* = 364)	Significant Stenosis (*n* = 182)	*p*-Value
Composite events	19 (4.5)	8 (3.0)	11 (7.4)	0.035
ACS	14 (2.6)	5 (1.4)	9 (4.9)	0.019
Cardiac death	1 (0.2)	1 (0.3)	0 (0.0)	1.000
VT or VF	2 (0.4)	1 (0.3)	1 (0.5)	1.000
AV block	2 (0.4)	1 (0.3)	1 (0.5)	1.000
All-cause death	2 (0.4)	2 (0.5)	0 (0.0)	0.555
**Total Period**	**All (*n* = 546)**	**No Significant Stenosis (*n* = 364)**	**Significant Stenosis (*n* = 182)**	***p*-Value**
Composite events	31 (5.7)	14 (3.8)	17 (9.3)	0.009
ACS	24 (4.4)	10 (2.7)	14 (7.7)	0.008
Cardiac death	2 (0.4)	2 (0.5)	0 (0.0)	0.555
VT or VF	4 (0.7)	2 (0.5)	2 (1.1)	0.604
AV block	2 (0.4)	1 (0.3)	1 (0.5)	1.000
All-cause death	3 (0.5)	3 (0.8)	0 (0.0)	0.554

ACS, acute coronary syndrome; AV, atrioventricular; VA, vasospastic angina; VF, ventricular fibrillation; VT, ventricular tachycardia.

**Table 5 jcm-10-03341-t005:** Predictors of 1-year ACS in patients with VA.

	Univariate			Multivariate		
	OR	95% CI	*p*-Value	OR	95% CI	*p*-Value
Significant coronary artery stenosis	5.36	2.335–12.303	<0.001	6.67	2.798–15.908	<0.001
Age	0.97	0.939–1.006	0.101	0.96	0.924–0.994	0.023
Male	0.91	0.398–2.060	0.813	1.20	0.514–2.820	0.670
Hypertension	0.91	0.401–2.077	0.828	-	-	-
Diabetes	0.84	0.193–3.581	0.811	-	-	-
Dyslipidemia	2.93	1.283–6.694	0.011	3.14	1.358–7.277	0.007
Current smoking	1.48	0.649–3.368	0.351	-	-	-
Alcohol drinking	0.95	0.422–2.114	0.890	-	-	-
Hemoglobin, g/dL	1.01	0.819–1.235	0.958	-	-	-
Creatinine, mg/dL	0.69	0.116–4.144	0.689	-	-	-
hs-CRP, mg/dL	1.02	0.959–1.075	0.599	-	-	-

ACS, acute coronary syndrome; CI, confidence interval; hs-CRP, high sensitive-C reactive protein; OR, odds ratio; VA, vasospastic angina.

## Data Availability

Data sharing not applicable.

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
