# Peer review of "Effect of Significant Coronary Artery Stenosis on Prognosis in Patients with Vasospastic Angina: A Propensity Score-Matched Analysis"

_jcm, 2021, doi:10.3390/jcm10153341_

Round 1

Reviewer 1 Report

Dear Authors,

It is a large registry based multicenter study with a presented follow up of one year  and not clearly two years.

A propensity score was used to adjust for differences in baseline characteristics between pts with vasospastic angina with significant (>50%) stenosis and no significant stenosis of coronary artery.

Authors found a difference of cumulative composite   event free survival and ACS free survival at one year on favor of pts without significant stenosis.

 My main critical remarks are:

1 No data were provided of the degree of stenosis, number of stenosed arteries in the stenosis group.

  1. There is no information whether any pts with significant stenoses of left main coronary artery or LAD were excluded from the analyses.
  2. No information was provided on the intervention (bypasses, PTCA, stents) – are those pts were excluded or included in the analyses? IF included this factor can influence the outcome and should be analyzed in multivariable analyses.
  3. Only data on admission medication were provided – no information on pharmacological therapy , after vasospasm was proven by provocative tests, was given, The type of medications is very important especially in the setting of pts with vasospastic angina. Do you found any differences between both analyzed groups?

5 In study outcomes 2.4 section you mentioned that you collected also data of two years follow-up. But no results of those prolonged follow –up were fully presented (if it is in table 3 it is not clear for the reader).

  1. I my opinion it is not a prospective study (see discussion) but a retrospective observational cohort study (see line 303)

Small remarks

  1. In Fig 1 B at the end of observation there is a large drop in of about 20% in ACS free survival in patients without stenosis without similar change in second (group without stenosis) . Is it true or it is a fault of presentation

Reviewer 2 Report

Dr Kim et al report on the effect of significant coronary artery stenosis on the prognosis in patients with vasospastic angina. They found that patients with significant CAD on top of VA have a poorer prognosis than those patients with only VA. While these results are neither entirely surprising or new, the manuscript is very well written and interesting to read! 

I have a few minor comments: 

  • page 2, line58/59: this sentence sounds like atherosclerosis in general is very harmless. Maybe rephrase to stable atherosclerotic plaques?
  • page 2, line 62: I think there is a word missing
  • page 2, line 70: patients, not patient
  • page 2, study population: I have not quite understood how the patients were selected. Did you include only patients who were referred with the specific question for VA or did you include also patients who were generally referred for an angiogram and who maybe had VA upon insertion of the catheter? Were the entered into the registry pre or post angio? The numbers in this paragraph are parts of the results and not the methods. 
  • page 2, data collection: how did you define diabetes mellitus, hypertension and dyslipidemia? Based on lab results, diagnoses, medications? (in the significant stenosis group, there are more patients with a statin than with a dyslipidemia -> why?), also, how did you deal with the previous CAD? What happened to those patients with significant stenosis? Did they undergo PTCA?
  • page 3, invasive CAG and EG provocation tests: it would be helpful if you state the most common indications. 
  • results, Table 3: it is not very well visible how this table is different from Table 2. Please add long-term-follow up. How long was long-term follow up?
  • page 7, clinical outcome: line 216 and 224: it would be great to state also absolute numbers and not only percentages. lines 222 - 224: sentence is difficult to understand, please rephrase. 
  • page 7, effect. It is interesting, how dyslipidemia is significant all of a sudden. How do you explain this? Is this between the two groups after propensity matching as well. It would be helpful to add the equivalent to Table1 with only those patients included to the appendix. 
  • Discussion: your second sentence is that there is no difference between single- and multi-vessel disease, however, this gets lost in the results. If you find this result important, elaborate in the results part. 
  • page 10, lines 293 ff. As mentioned before, this is a bit strange. Not clearly defined in the methods, not significant in Table1 and suddenly it appears to be very important. This needs some explanation. 
  • I think the article by Ishii et al would be important to mention in the discussion. 
    • PMID: 26337988
    • DOI: 10.1016/j.jacc.2015.06.1324

Round 2

Reviewer 1 Report

Dear Authors

Thank you for your explanations and corrections

With best regards